# Visual Perceptual Quality Assessment Based on Blind Machine Learning Techniques

**DOI:** 10.3390/s22010175

**Published:** 2021-12-28

**Authors:** Ghislain Takam Tchendjou, Emmanuel Simeu

**Affiliations:** Univ. Grenoble Alpes, CNRS, Grenoble INP (Institute of Engineering Univ. Grenoble Alpes), TIMA, 38000 Grenoble, France

**Keywords:** blind image quality assessment, regression technique, non-distortion-specific, visual perception, convolutional neural network, deep learning, *ILBQA*, *FPGA* implementation

## Abstract

This paper presents the construction of a new objective method for estimation of visual perceiving quality. The proposal provides an assessment of image quality without the need for a reference image or a specific distortion assumption. Two main processes have been used to build our models: The first one uses deep learning with a convolutional neural network process, without any preprocessing. The second objective visual quality is computed by pooling several image features extracted from different concepts: the natural scene statistic in the spatial domain, the gradient magnitude, the Laplacian of Gaussian, as well as the spectral and spatial entropies. The features extracted from the image file are used as the input of machine learning techniques to build the models that are used to estimate the visual quality level of any image. For the machine learning training phase, two main processes are proposed: The first proposed process consists of a direct learning using all the selected features in only one training phase, named direct learning blind visual quality assessment DLBQA. The second process is an indirect learning and consists of two training phases, named indirect learning blind visual quality assessment ILBQA. This second process includes an additional phase of construction of intermediary metrics used for the construction of the prediction model. The produced models are evaluated on many benchmarks image databases as TID2013, LIVE, and LIVE in the wild image quality challenge. The experimental results demonstrate that the proposed models produce the best visual perception quality prediction, compared to the state-of-the-art models. The proposed models have been implemented on an FPGA platform to demonstrate the feasibility of integrating the proposed solution on an image sensor.

## 1. Introduction

Digital images are increasingly used in several vision application domains of everyday life, such as medical imaging [1,2], object recognition in images [3], autonomous vehicles [4], Internet of Things (IoT) [5], computer-aided diagnosis [6], and 3D mapping [7]. In all these applications, the produced images are subject to a wide variety of distortions during acquisition, compression, transmission, storage, and displaying. These distortions lead to a degradation of visual quality [8]. The increasing demand for images in a wide variety of applications involves perpetual improvement of the quality of the used images. As each domain has different thresholds in terms of visual perception needed and fault tolerance, so, equally, does the importance of visual perception quality assessment.

As human beings are the final users and interpreters of image processing, subjective methods based on the human ranking score are the best processes to evaluate image quality. The ranking consists of asking several people to watch images and rate their quality. In practice, subjective methods are generally too expensive, time-consuming, and not usable in real-time applications [8,9]. Thus, many research works have focused on objective image quality assessment IQA methods aiming to develop quantitative measures that automatically predict image quality [2,8,10]. The objective IQA process is illustrated in Figure 1. This process has been introduced in the paper [11].

In digital image processing, objective IQA can be used for several roles, such as dynamic monitoring and adjusting the image quality, benchmark and optimize image processing algorithms, and parameter setting of image processing [8,12]. Many research investigations explore the use of machine learning (ML) algorithms in order to build objective IQA models in agreement with human visual perception. Recent methods include, artificial neural network (ANN), support vector machine (SVM), nonlinear regression (NLR), decision tree (DT), clustering, and fuzzy logic (FL) [13]. After the propulsion of deep learning techniques in 2012 [14], researchers were also interested in the possibility of using these techniques in the image quality assessment. Thus, in 2014, the first studies emerged on the use of convolutional neural networks in IQA [15]. Many other works have followed, and we find in the literature increasingly efficient IQA models [16,17,18].

The IQA evaluation methods can be classified into three categories according to whether or not they require a reference image: full-reference (FR), reduced-reference (RR), and no-reference (NR) IQA approaches. Full reference image quality assessment (FR-IQA) needs a complete reference image in order to be computed. Among the most popular FR-IQA, we can cite the peak signal to noise ratio (PSNR), structure similarity index metric (SSIM) [8,19], and visual information fidelity (VIF) [20]. In reduced reference image quality assessment (RR-IQA), the reference image is only partially available, in the form of a set of extracted features, which help to evaluate the distorted image quality; this is the case of reduced reference entropic differencing (RRED) [21]. In many real-life applications, the reference image is unfortunately not available. Therefore, for this application, the need of no-reference image quality assessment (NR-IQA) methods or blind IQA (BIQA), which automatically predict the perceived quality of distorted images, without any knowledge of reference image. Some NR-IQA methods assume the type of distortions are previously known [22,23], these objective assessment techniques are called distortion specific (DS) NR-IQA. They can be used to assess the quality of images distorted by some particular distortion types. As example, the algorithm in [23] is for JPEG compressed images, while in [22] it is for JPEG2000 compressed images, and in [24] it is for detection of blur distortion. However, in most practical applications, information about the type of distortion is not available. Therefore, it is more relevant to design non-distortion specific (NDS) NR-IQA methods that examine image without prior knowledge of specific distortions [25]. Many existing metrics are the base units used in NDS methods, such as DIIVINE [26], IL-NIQE [27], BRISQUE [28], GMLOGQA [29], SSEQ [30], FRIQUEE [31], DIQA [32], and DIQa-NR [33].

The proposal of this paper is a non-distortion-specific NR-IQA approach, where the extracted features are based on a combination of the natural scene statistic in the spatial domain [28], the gradient magnitude [29], the Laplacian of Gaussian [29], as well as the spatial and spectral entropies [30]. These features are trained using machine learning methods to construct the models used to predict the perceived image quality. The process we propose for designing the evaluation of perceived no-reference image quality models is described in Figure 2. The process consists of

extracting the features from images taken in the IQA databases,removing the superfluous features according to the correlation between the extracted features,grouping the linearly independent features to construct some intermediate metrics, andusing the produced metrics to construct the estimator model for perceived image quality assessment.

Finally, we compare the designed models with the state-of-the-art models using the features extraction, but also the process using deep learning with convolutional neural network; as shown in Section 4.1.

To evaluate the performances of the produced models, we measure the correlation between objective and subjective quality scores using three correlation coefficients:(1)Pearson’s linear correlation coefficient (PLCC), which is used to measure the degree of the relationship between linear related variables.(2)Spearman’s rank order correlation coefficient (SRCC), which is used to measure the prediction monotony and the degree of association between two variables.(3)Brownian distance (dCor), which is a measure of statistical dependence between two random variables or two random vectors of arbitrary, not necessarily equal dimension.

The paper is organized as follows. Section 2 presents the feature extraction methods. Section 3 explains the feature selection technique based on feature independence analysis. The construction of intermediate metrics is also presented in this section. Section 4 explains the experimental results and their comparison. Section 5 presents the FPGA implementation architectures and results. Finally, Section 6 draws a conclusion and perspectives for future investigations.

## 2. Feature Extraction

Feature extraction in this paper is based on four principal axes: natural scene statistic in the spatial domain, gradient magnitude, Laplacian of Gaussian, and finally spatial and spectral entropies.

### 2.1. Natural Scene Statistic in the Spatial Domain

The extraction of the features based on NSS in spatial domain starts by normalization of the image represented by I(i,j), to remove local mean displacements from zero log-contrast, and to normalize the local variance of the log contrast as observe in [34]. Equation (Equation 1) presents normalization of the initial image.
(1)$$\hat{I}\left(i,j\right)=\frac{I(i,j)-\mu (i,j)}{\lambda (i,j)+1}$$
where *i* and *j* are the spatial indices, M and N are the image dimensions, i∈1,2,…,M and j∈1,2,…,N. μ(i,j) that denotes the local mean is represented by (Equation 2) and λ(i,j) that estimates the local contract is expressed by (Equation 3).
(2)$$\mu \left(i,j\right)=\sum_{k}\sum_{l}\omega_{k,l}I\left(i+k,j+l\right)$$
(3)$$\lambda \left(i,j\right)=\sqrt{\sum_{k}\sum_{l}\omega_{k,l}{\left[I\left(i+k,j+l\right)-\mu \left(i,j\right)\right]}^{2}}$$
where k=−K,…,K, and l=−L,…,L. $\omega_{k,l}$ is a 2D circularly-symmetric Gaussian weighting function sampled out to 3 standard deviations (K=L=3) and rescaled to unit volume [28].

The model produced in (Equation 1) is used as the mean-subtracted contrast normalized (MSCN) coefficients. In [28], they take the hypothesis that the MSCN coefficients have characteristic statistical properties that are changed by the presence of distortion. Quantifying these changes helps predict the type of distortion affecting an image as well as its perceptual quality. They also found that a Generalized Gaussian Distribution (GGD) could be used to effectively capture a broader spectrum of a distorted image, where the GGD with zero mean is given by (Equation 4).
(4)$$f\left(x;\alpha ,\sigma^{2}\right)=\frac{\alpha }{2\beta \Gamma (1/\alpha )}\exp\left(-{\left(x/\beta \right)}^{\alpha }\right)$$
where β is represented by (Equation 5) and Γ(.) is expressed by (Equation 6).
(5)$$\beta =\sigma \sqrt{\frac{\Gamma \left(1/\alpha \right)}{\Gamma \left(3/\alpha \right)}}$$
(6)$$\Gamma \left(a\right)=\int_{0}^{\infty }t^{a-1}e^{-t}dt\;\;\;a>0.$$

The parameter α controls the shape of the distribution while $\sigma^{2}$ controls the variance.

In [28], they also give the statistical relationships between neighboring pixels along four orientations: $H=\hat{I}\left(i,j\right)\hat{I}\left(i,j+1\right);V=\hat{I}\left(i,j\right)\hat{I}\left(i+1,j\right);D1=\hat{I}\left(i,j\right)\hat{I}\left(i+1,j+1\right)$ and $D2=\hat{I}\left(i,j\right)\hat{I}\left(i+1,j-1\right)$. This is used with asymmetric density function to produce a practical alternative to adopt a general asymmetric generalized Gaussian distribution (AGGD) model [35]. Equation (Equation 7) gives the AGGD with zero mode.
(7)$$f\left(x;\gamma ,\sigma_{l}^{2},\sigma_{r}^{2}\right)=\left\{\begin{array}{c} \frac{\gamma }{\left(\beta_{l}+\beta_{r}\right)\Gamma \left(1/\gamma \right)}\exp\left(-{\left(\frac{-x}{\beta_{l}}\right)}^{\gamma }\right)\forall x\leq 0\\ \frac{\gamma }{\left(\beta_{l}+\beta_{r}\right)\Gamma \left(1/\gamma \right)}\exp\left(-{\left(\frac{x}{\beta_{r}}\right)}^{\gamma }\right)\forall x\geq 0 \end{array}\right.$$
where $\beta_{a}$ (with a=lorr) is given by (Equation 8).
(8)$$\beta_{a}=\sigma_{a}\sqrt{\frac{\Gamma \left(1/\gamma \right)}{\Gamma \left(3/\gamma \right)}}$$

The parameter γ controls the shape of the distribution, while $\sigma_{l}^{2}$ and $\sigma_{r}^{2}$ are scale parameters that control the spread on each side of the mode. The fourth asymmetric parameter is η given by (Equation 9)
(9)$$\eta =\left(\beta_{r}-\beta_{l}\right)\frac{\Gamma \left(2/\gamma \right)}{\Gamma \left(1/\gamma \right)}$$

Finally, the founded parameters are composed of the symmetric parameters (α and $\sigma^{2}$) and the asymmetric parameters ($\eta ,\gamma ,\sigma_{r}^{2}$, and $\sigma_{l}^{2}$), where the asymmetric parameters are computed for the four orientations, as shown in Table 1. All the founded parameters are also computed for two scales, yielding 36 features (2 scales × [2 symmetric parameters +4 asymmetric parameters ×4 orientations]). More details about the estimation of these parameters are given in [28,36].

### 2.2. Gradient Magnitude and Laplacian of Gaussian

The second feature extraction method is based on the joint statistics of the Gradient Magnitude (GM) and the Laplacian of Gaussian (LoG) contrast. These two elements GM and LoG are usually used to get the semantic structure of an image. In [29], they also introduce another usage of these elements as features to predict local image quality.

By taking an image *I*(*i*,*j*), its GM is represented by (Equation 10).
(10)$$G_{I}=\sqrt{{\left[I⊗h_{x}\right]}^{2}+{\left[I⊗h_{y}\right]}^{2}}$$
where ⊗ is the linear convolution operator, and $h_{d}$ is the Gaussian partial derivative filter applied along the direction d∈x,y, represented by (Equation 11).
(11)$$h_{d}\left(x,y\sigma \right)=-\frac{1}{2\pi \sigma^{2}}\frac{d}{\sigma^{2}}\exp\left(-\frac{x^{2}+y^{2}}{2\sigma^{2}}\right)$$

Moreover, the LoG of this image is represented by (Equation 12).
(12)$$L_{I}=I⊗h_{LoG}$$
where
(13)$$h_{LoG}\left(x,y\sigma \right)=-\frac{1}{2\pi \sigma^{2}}\frac{x^{2}+y^{2}-2\sigma^{2}}{\sigma^{4}}\exp\left(-\frac{x^{2}+y^{2}}{2\sigma^{2}}\right)$$

To produce the used features, the first step is to normalize the GM and LoG features map as in (Equation 14).
(14)$$\begin{array}{c} {\bar{G}}_{I}=G_{I}/\left(N_{I}+\epsilon \right)\\ {\bar{L}}_{I}=L_{I}/\left(N_{I}+\epsilon \right) \end{array}$$
where ϵ is a small positive constant, used to avoid instabilities when $N_{I}$ is small, and $N_{I}$ is given by (Equation 15).
(15)$$N_{I}\left(i,j\right)=\sqrt{\sum_{l}\sum_{k}\omega \left(l,k\right)F_{I}^{2}\left(l,k\right)}$$
where $\left(l,k\right)\in \Omega_{i,j}$; and $F_{I}$ is given by (Equation 16).
(16)$$F_{I}\left(i,j\right)=\sqrt{G_{I}^{2}\left(i,j\right)+L_{I}^{2}\left(i,j\right)}$$

Then (Equation 17) and (Equation 18) give the final statistic features.
(17)$$\left\{\begin{array}{c} P_{G}\left(G=g_{m}\right)=\sum_{n}K_{m,n}\\ P_{L}\left(L=l_{n}\right)=\sum_{m}K_{m,n} \end{array}\right.$$
(18)$$\left\{\begin{array}{c} Q_{G}\left(G=g_{m}\right)=\frac{1}{N}\sum_{n}P\left(G=g_{m}L=l_{n}\right)\\ Q_{L}\left(L=l_{n}\right)=\frac{1}{M}\sum_{m}P\left(L=l_{n}G=g_{m}\right) \end{array}\right.$$
where n=1,…,N and m=1,…,M, and $K_{m,n}$ is the empirical probability function of G and L [37,38]; it can be given by (Equation 19).
(19)$$K_{m,n}=P\left(G=g_{m},L=l_{n}\right)$$

In [29], the authors also found that the best results are obtained by setting M=N=10; thus, 40 statistical features have been produced as shown in Table 2, 10 dimensions for each statistical features vector $P_{G},\;P_{L},\;Q_{G}$ and $Q_{L}$.

### 2.3. Spatial and Spectral Entropies

Spatial entropy is a function of the probability distribution of the local pixel values, while spectral entropy is a function of the probability distribution of the local discrete cosine transform (DCT) coefficient values. The process of extracting the spatial and spectral entropies (SSE) features from images in [30] consists of three steps:The first step is to decompose the image into 3 scales, using bi-cubic interpolation: low, middle, and high.The second step is to partition each scale of the image into 8×8 blocks and compute the spatial and spectral entropies within each block. The spatial entropy is given by (Equation 20).
(20)$$E_{s}=-\sum_{x}p\left(x\right)log_{2}p\left(x\right)$$
and spectral entropy is given by (Equation 21).
(21)$$E_{f}=-\sum_{i}\sum_{j}P\left(i,j\right)log_{2}P\left(i,j\right).$$
where p(x) is the probability of x, and P(i,j) is the spectral probability gives by (Equation 22).
(22)$$P\left(i,j\right)=\frac{C{\left(i,j\right)}^{2}}{\sum_{i}\sum_{j}C{\left(i,j\right)}^{2}}.$$
where i=1,…,8;j=1,…,8.In the third step, evaluate the means and skew of blocks entropy within each scales.

At the end of the three steps, 12 features are extracted from the images as seen in Table 3. These features represent the mean and skew for spectral and spatial entropies, on 3 scales (2×2×3=12 features).

### 2.4. Convolutional Neural Network for NR-IQA

In this paper, we explore the possibility of use the deep learning with convolutional neural network to build the model used to evaluate the quality of the image. In this process, the extraction of features are done by the convolution matrix, constructed using the training process with deep learning.

In CNNs, three main characteristics of convolutional layers can distinguish them from fully connected linear layers in the vision field. In the convolutional layer,

each neuron receives an image as inputs and produces an image as its output (instead of a scalar);each synapse learns a small array of weights, which is the size of the convolutional window; andeach pixel in the output image is created by the sum of the convolutions between all synapse weights and the corresponding images.

The convolutional layer takes as input and image of dimension $R_{l}xC_{l}$ with $S_{l}$ channels, and the output value of the pixel $p_{l,n,r,c}$ (the pixel of the row *r* and the column *c*, of the neuron *n* in the layer *l*) is computed by (Equation 23).
(23)$$p_{l,n,r,c}=f_{act}\left(\sum_{s=0}^{S_{l}}\sum_{i=0}^{I_{l}}\sum_{j=0}^{J_{l}}w_{l,n,s,i,j}p_{l-1,n,r+i,c+j}\right)$$
where $I_{l}\times J_{l}$ are the convolution kernel dimensions of the layer *l*, $w_{l,n,s,j,k}$ is the weight of the row *i* and the column *j* in the convolution matrix of the synapse *s*, connected to the input of the neuron *n*, in the layer *l*. In reality, a convolution is simply an element-wise multiplication of two matrices followed by a sum. Therefore, the 2D convolution take two matrices (which both have the same dimensions), multiply them, element-by-element, and sum the elements together. To close the convolution process in a convolutional layer, the results are then passed through an activation function.

In convolution process, a limitation of the output of the feature map is that they record the precise position of features in the input in the convolutional layers. This means that small movements in the position of the feature in the input image will result in a different feature map. This can happen with cropping, rotating, shifting, and other minor changes to the input image [15]. In CNNs, the common approach to solving this problem is the down-sampling using the pooling layers [39]. The down-sampling is a reduction in the resolution of an input signal while preserving important structural elements, without the fine details that are not very useful for the task. The pooling layers are used to reduce the dimensions of the feature maps. Thus, it reduces the number of parameters to learn and the amount of computation performed in the network. The pooling layer summarizes the features present in a region of the feature map generated by a convolution layer. Therefore, further operations are performed on summarized features instead of precisely positioned features generated by the convolution layer. This makes the model more robust to variations in the position of the features in the input image.

In CNNs, the most used pooling function is Max Pooling, which calculates the maximum value for each patch on the feature map. But other pooling function exist, like Average Pooling, which calculates the average value for each patch on the feature map.

Our final CNN model, called CNN-BQA has 10 layers: four convolutional layers with 128, 256, 128, and 64 channels, respectively; four max pooling layers; and two fully connected layers with 1024 and 512 neurons, respectively. Finally, a output layer with one neuron is computed to give the final score of the image.

## 3. Construction of Prediction Models

In the feature extraction phase, 88 features were extracted in the images. These features are used to construct the prediction models used to evaluate the quality of images. In our case, two processes have been proposed to evaluate the image quality:The Direct Learning Blind visual Quality Assessment method (DLBQA), in which machine learning methods are directly applied overall set of selected features for training, and producing the final models.Indirect Learning Blind visual Quality Assessment method (ILBQA), in which ML methods are applied to the four produced intermediary metrics to construct the final model. The ILBQA process requires two training phases: the first is used to merge the independent features of each class in order to generate the adequate intermediary metrics, while the task of the second training is to derive the final IQA model. The inputs of the produced model are the four produced intermediary metrics, and the output is the estimated quality score of the input image.

### 3.1. Remove Superfluous Features

Having extracted the 88 features on the images, the next step consists of removing the superfluous features, which not add any useful information for the construction of final IQA models. The main type of superfluous features consists of strongly correlated features.

The strongly correlated features give approximately the same information, and this redundancy may cause overfitting problem in the estimation process of the model parameters, and can also produce uselessly complex final models. The proposed way in this paper in order to avoid these problems is to remove the strongly correlated features. The suppression of features is based on the following idea: “if there are two strongly correlated features in the set of extracted features, with a mutual correlation greater than a predefined threshold, we remove the one that has the least correlation with the subjective score without significant loss of information”. This reduction process is clearly described in Figure 3, which takes as input the set of extracted features, and produces the set of superfluous features.

The features contained in the output vector of this algorithm are removed from the list of those to be used in the rest of the work. In our use case, we took 0.9 (thres=90%) as the threshold of mutual correlation between two features. This implies that for any mutual correlation between 2 features higher than 0.9, one of the two features concerned must be removed. With this threshold, the output vector contains 42 features to remove from the initial extracted features list. At the end of this step, 42 features have been found as superfluous features; yielding 46 features are saved for study.

### 3.2. Direct Process (DLBQA)

In the direct process, the whole set of 46 selected features is used by only one training phase to produce the final quality assessment model. The produced models have been called Direct Learning Blind visual Quality Assessment index based on Machine Learning (DLBQA-ML). Different machine learning methods have been used to construct assessment models. These include methods based on artificial neural networks (DLBQA-ANN), polynomial Nonlinear Regression (DLBQA-NLR), Support Vector Machine (DLBQA-SVM), Decision Tree (DLBQA-DT), and Fuzzy Logic (DLBQA-FL). This different machine learning methods have been presented in [11,40]

Table 4 resumes the results using different correlation scores on the test data by the DLBQA-ML models produced using different machine learning methods, on the TID2013 [41,42], LIVE [8,43,44], and LIVE challenge [45,46] image databases.

These results show that the DLBQA based on support vector machine (SVM) and decision tree (DT) methods provide the best models for predicting image quality from the different image quality databases.

### 3.3. Indirect Process (ILBQA)

As the number of selected features is too big (46), the indirect process devises the training phase into three steps:The first step, as mentioned in the previous section, is to distribute the features into independent classes, depending on the axes of the extracted features.The next step uses machine learning methods to merge the features of each class in order to generate an appropriate intermediary metrics. Then, the pool process has been carried out using different ML methods to create intermediary metrics IM_NSS, IM_GM, IM_LOG, and IM_SSE.The third step consists of using the produced intermediary metrics to construct final model and to derive the quality score estimators.

#### 3.3.1. Construction of Intermediate Models

Having removed superfluous features, the next step consists of grouping the selected features into a set of classes. The produced classes are based on four principal axes:Natural Scene Statistic in the spatial domain (NSS) with 14 extracted features;Gradient Magnitude (GM) with 13 extracted features;Laplacian of Gaussian (LOG) with 11 extracted features;Spatial and Spectral Entropies (SSE) with 8 extracted features.

Each produced class is used to construct one intermediary metric. In each produced class, the features have been merged to generate the appropriate intermediary metrics. The pool process has been carried out, using different machine learning methods.

Table 5 compares the result of applying four ML methods (fuzzy logic, support vector machine, decision tree, and artificial neural network) in order to build the intermediary metrics, using the feature classes. The comparison is represented by Spearman’s correlation score between the produced intermediary metrics and the subjective scores give in different image databases.

The selection of the adequate ML method is determined by studying Brownian correlations between the produced intermediary metrics and the subjective scores in multivariate space as shown in Table 6.

The main remark extracts on this Table 6 is that the decision tree (DT) ML method produces the best results on all the studied image databases. Thus, the decision tree method has been used as the machine learning method to learn the intermediary metrics.

#### 3.3.2. Construction of Final Indirect Prediction Model

Once the intermediary metrics are built, the next step in the indirect process is the construction of final prediction model using these intermediary metrics. The final evaluation index constructed consists of several models: four models of construction of the intermediary metrics, and a model of evaluation of the final score using the intermediary metrics. As the intermediary metrics have been constructed in the previous section, all that remains is the construction of the final prediction model, taking as inputs these intermediary metrics previously evaluated. These models are built using different machine learning methods.

Table 7 resumes the prediction performance of ILBQA-based image quality models trained on TID213, LIVE, and Live challenge image databases. The design of IQA models consists here of two training steps. The first step uses the decision tree approach in order to merge the quality features into four intermediary metrics. Moreover, the next training step is used different ML approaches (DT, FL, SVM, and ANN) for predicting quality score from the corresponding set of intermediary metrics.

These results show that the ILBQA based on decision tree (DT) methods provide the best models for predicting image quality from the different image quality databases.

## 4. Experimental Results

Having extracted and removed the superfluous features, the next step is to construct machine learning models, which automatically predict the perceived quality of an input image without a need to reference image. The construction process of the objective NR-IQA block essentially consists of training (with α% of data) and validation phases. During the training phase, machine learning algorithms are applied to create an appropriate mapping model that relates the features extracted from an image to subjective score provided in the image database. While the validation phase is used to estimate how well the trained ML models predict the perceived image quality, and to compare the accuracy and robustness of the different ML methods.

The Monte Carlo Cross-Validation (MCCV) method [47] is also applied to compare and confirm the accuracy of the obtained results, as follows:First, split the dataset with a fixed fraction (α%) into training and validation set, where α% of the samples are randomly assigned to the training process in order to design a model, and the remaining samples are used for assessing the predictive accuracy of the model.Then, the first step is repeated k times.Finally, the correlation scores are averaged over k.

In this paper, α%=70% and k=1000.

### 4.1. Results Comparison

Having built our evaluation models, the next step is the validation of these models, and the comparison of the results produced with those produced by the state-of-the-art models. The validation process is done using Monte Carlo Cross-Validation (MCCV) [47].

In Figure 4, Figure 5 and Figure 6, different state-of-the-art no-reference IQA methods have been studied and compared with the two models design in this paper (DLBQA and ILBQA). These blind IQA methods consist of BRISQUE [28] that uses NSS in the spatial domain, GMLOGQA [29] that uses the gradient magnitude (GM) and the Laplacian of Gaussian (LOG) to make a predictor for image quality, and SSEQ [30] based on the spatial and spectral entropies. These figures depict the box-plot of Brownian’s distance in the MCCV phase, of different no-reference IQA methods, trained on TID2013, LIVE, and LIVE Challenge image databases, respectively. They also show that the BIQA model that obtains the best prediction accuracy (with the highest mean and the smallest standard deviation) is designed by the proposed method ILBQA with two training steps.

We compare the proposed models (DLBQA, ILBQA, and CNN-BQA) with six feature extraction based BIQA methods, and five CNN-based methods, on three popular benchmark IQA databases. The results are listed in Table 8. In the table, the best two SROCC and PLCC are highlighted in bold. The performances are the correlations between the subjective score (MOS/DMOS) take in image databases, and the evaluated objective scores given by the different studied NR-IQA methods in cross-validation phase with k=1000. The main remark is that for every image databases (TID2013, LIVE, and LIVE Challenge), the proposed DLBQA and ILBQA methods perform the best results.

### 4.2. Computational Complexity

In many image quality assessment applications, the computational complexity is an important factor when evaluating the produced BIQA models. In this paper, the feature extraction is based on four principal axes. Table 9 presents the percentage of time spent by each feature extraction axes.

The run-time of the produced model is compared to the run-times of five other good NR-IQA indexes: BLIINDS-II [50], DIIVINE [26], BRISQUE [28,36], GMLOGQA [29], and SSEQ [30]. Table 10 presents the estimated run-time for an execution on CPU, in second(s), of the different methods, for a BMP image with resolution 512×768, on MATLAB, using a computer with 16GB RAM and Intel Xeon dual-core CPU, 1.70 GHz for each core. We observe that the run-time of the proposed method (ILBQA) is significantly inferior to DIIVINE and BLIIND-II run-time, approximately equal to SSEQ run-time, and superior to BRISQUE and GMLOGQA run-time.

## 5. Implementation Architectures and Results

After the construction of the no-reference image quality evaluation models, through the MATLAB tool, the next step in our process is the implementation of the produced models on an FPGA platform. The produced IP-Core can be integrated into the output of an image sensor to evaluate the quality of the produced images. The implemented index is the indirect process (ILBQA). This implementation is done with the Xilinx Vivado and Vivado HLS and is implemented on the Xilinx Virtex 7 (VC707) and Zynq 7000 (XC7Z020) platforms.

The work in [11] introduces this phase of implementation of our proposed process on the objective no-reference image quality assessment. The general implementation process consists of the following.

HLS implementation of the block IP, allowing the encapsulation of the designed models, through the Vivado HLS tools, in C/C++ codes.Integration of the produced IP-cores in the overall design at RTL, through the Vivado tool, in VHDL/Verilog codes.Integration on Xilinx’s Virtex 7 (VC707) and Zynq 7000 (XC7Z020) FPGA boards, and evaluation of the obtained results on several test images.

### 5.1. HLS Implementation of the IP-Cores

The implementation of the ILBQA models in HLS enabled the construction of five IP-cores. This IP-Cores are divided into three groups, leading to the evaluation of the objective score of an image taken at the input of the process. The IP-Cores in each group can run in parallel, but the inputs in each group depend on the outputs of the previous group. The three proposed groups consist of feature extraction, construction of intermediate metrics based on decision tree models, and evaluation of the objective score from the model based on fuzzy logic, taking as input the intermediate metrics. The layered architecture of the produced IP-cores is presented in Figure 7.

Feature extraction: the entries of this group are, the image whose quality is sought to be evaluated, as an AXI-stream, and the dimensions of this input image (row and column sizes as an integer). It contains three IP-cores: NSS_Core which extracts 14 features on the image; GM-LOG_Core consisting of extraction of 13 and 11 features on GM and LoG axes, respectively; and SSE_Core which extracts 08 features.Production of the intermediary metrics: This group contains only one IP-core, the decision tree (DT) IP-core. The DT_Core uses the features extracted from the input image and generates the intermediary metrics in floating-point format.Production of the estimated score of the image using the intermediary metrics: this group also has only one IP-core, fuzzy logic IP-core (FL_Core). This IP-core takes as input the four intermediary metrics produced by the DT_Core and produces the estimated image quality score of the input image, in floating-point format.

Table 11 presents the estimation of performances and utilization at the synthesis level mapped to Xilinx Virtex 7 (VC707) FPGA platform, for the different produced IP-cores, and the global process. These results are produced by an image with a resolution of 512×768, and for a clock cycle equal to 25 ns (a frequency equal to 40 Mhz).

### 5.2. SDK Implementation and Results

The global design containing our models has been implemented on an FPGA board, then a program is implemented in the micro-controller (MicroBlaze) to evaluate the quality of the image thus received through the model implemented on FPGA. This program also return the objective score evaluated by the FPGA implemented models. The images are retrieved in “AXI-Stream” format and transmitted on the FPGA board through an extension using the HDMI protocol. The program is written in C/C++ codes, and passed to micro-controller via the JTAG interface of the FPGA board, thanks to Xilinx SDK tool.

120 images distorted by 24 types of distortion, from the TID2013 image database [41,42] were evaluated by the produced models on the FPGA platform. The FPGA implementation results were compared to the experimental results under MATLAB and also to the subjective scores taken the image database.

Figure 8 presents the comparison between MATLAB results in the X-axis and FPGA implementation results in the Y-axis, for the ILBQA index. While Figure 9 presents the comparison between the subjective scores in the X-axis, and MATLAB results (in blue) and FPGA implementation results (in red) in the Y-axis. These figures also present the correlation scores between the scores in the X-axis and the produced objective scores in the Y-axis on each graph.

The difference between the image quality scores estimated under MATLAB and those estimated on FPGA is caused by several factors, including the following.

Spreading 32-bit floating-point truncation errors in the FPGA system;The difference between the results of the basic functions of the MATLAB and Vivado HLS tools. For example, the conversion functions in gray-scale, or the resizing functions using the bi-cubic interpolation, give slightly different results on MATLAB and Vivado HLS. These small errors are propagated throughout the evaluations, to finally cause the differences between the estimated scores shown in the previous figures.

### 5.3. Visual Assessment

Our NR-IQA process has been used in a control loop process [51] to assess the quality of an image and decide if that image needs correction. Gaussian noise is inserted in a reference image, resulting in distorted image illustrated in Figure 10a, with a very low ILBQA score like show in the figure. Figure 10b–d shows the corrected version of the distorted image presented in the graph (a), by the control loop process based on the image quality assessment. Each graph also presents the PSNR between the reference image and the corrected image, as well as the blind image quality (ILBQA) score of each image. The graphs (b) and (c) of the Figure 10 show the intermediary corrected images, while the graph (d) presents the best-corrected image by the process.

## 6. Conclusions

This paper has presented the construction of models for objective assessment of perceived visual quality. The proposed models use a non-distortion specific no-reference image quality assessment methods. The objective IQA is computed by combining the most significant image features extracted from natural scene statistics (NSS) in the spatial domain, gradient magnitude (GM), Laplacian of Gaussian (LoG), and spectral and spatial entropies (SSE). In our proposal, the training phase has been performed using DLBQA and ILBQA process. The DLBQA index evaluates the image quality using all the selected features in one training phase. While the ILBQA index uses two training phases. The first phase consists of training the intermediary metrics using the feature classes based on four extraction axes, and the second training phase evaluates the image quality using the intermediary metrics. Different machine learning methods were used to assess image quality and their performance was compared. The proposed methods consisting of artificial neural network (ANN), nonlinear polynomial regression (NLR), decision tree (DT), support vector machine (SVM), and fuzzy logic (FL) are discussed and compared. Both the stability and the robustness of designed models are evaluated using a variant of Monte Carlo cross-validation (MCCV) with 1000 randomly chosen validation image subsets. The accuracy of the produced models has been compared to the results produced by other state-of-the-art NR-IQA methods. Implementation results on MATLAB and on the FPGA platform demonstrated the best performances for ILBQA proposed in this paper, which uses two training phases in the modeling process. One of the future work based on this article is to present how to use our image quality assessment indexes in the control loop process for self-healing of the image sensor. In addition, the implementation on an ASIC and the implementation of the generated IP-core in image sensors.

## Figures and Tables

**Figure 1 sensors-22-00175-f001:**
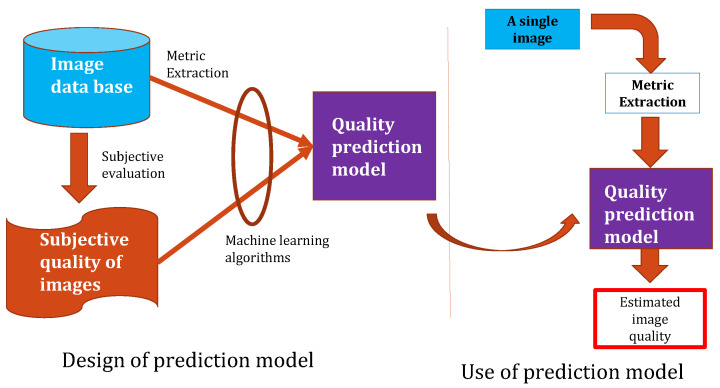
Objective image quality assessment process.

**Figure 2 sensors-22-00175-f002:**
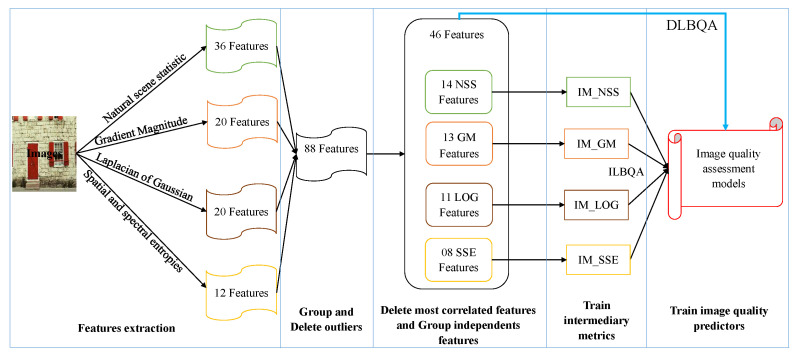
Design of prediction models for objective no-reference IQA process.

**Figure 3 sensors-22-00175-f003:**
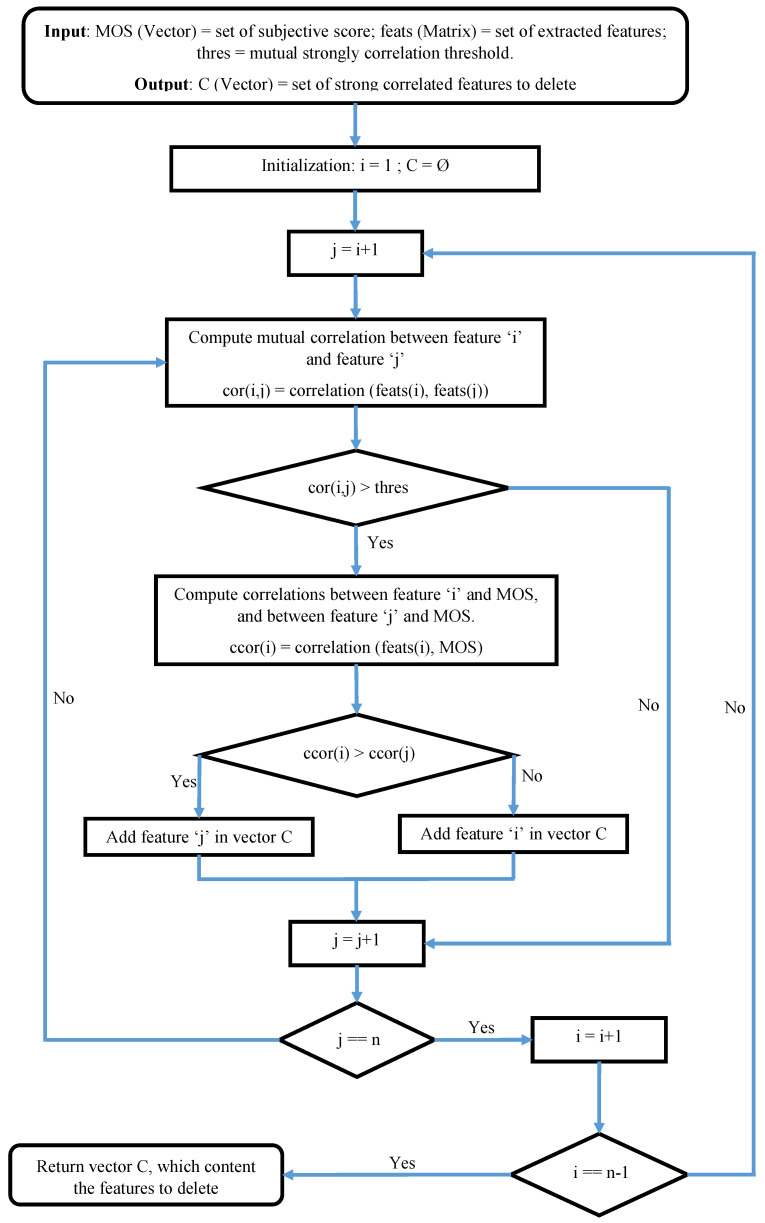
Flow diagram for selection of strongly correlated features to remove.

**Figure 4 sensors-22-00175-f004:**
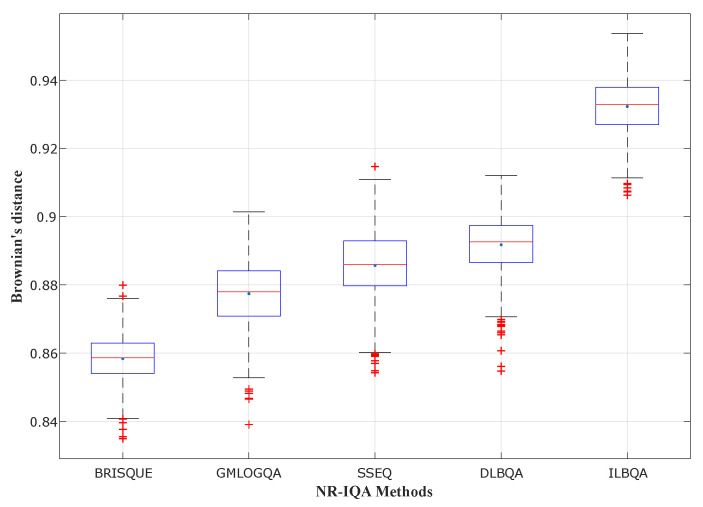
Performance comparison for different NR-IQA methods trained on TID2013.

**Figure 5 sensors-22-00175-f005:**
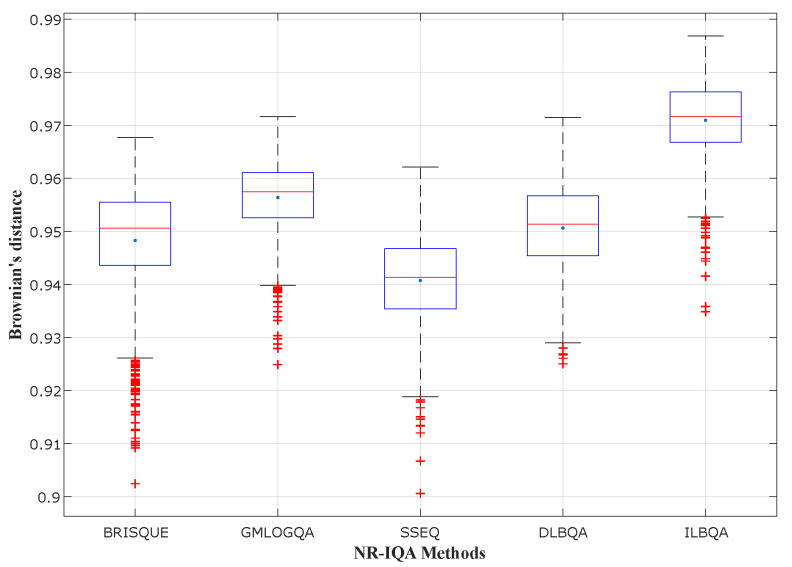
Performance comparison for different NR-IQA methods trained on LIVE.

**Figure 6 sensors-22-00175-f006:**
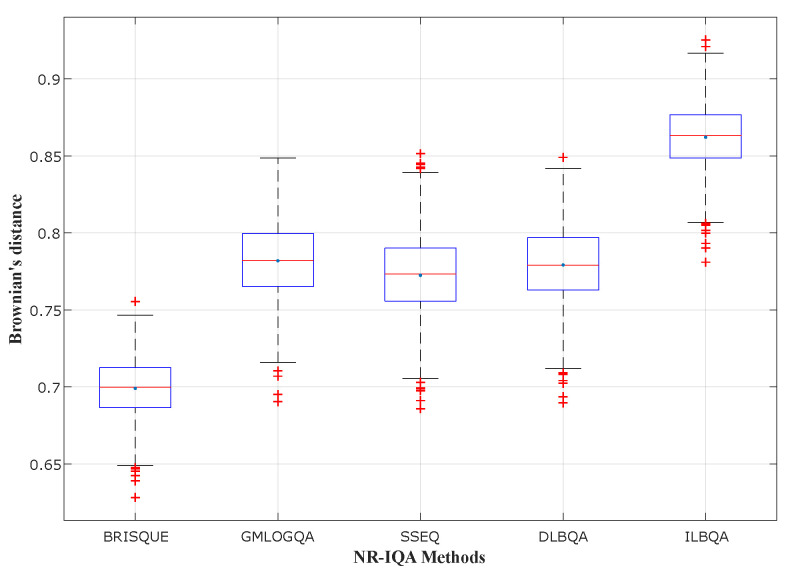
Performance comparison for different NR-IQA methods trained on LIVE Challenge.

**Figure 7 sensors-22-00175-f007:**
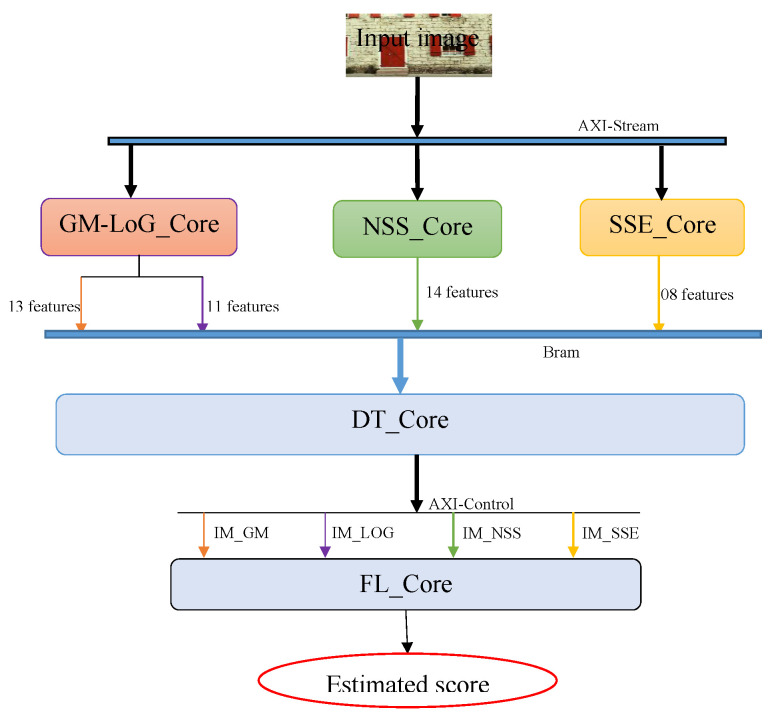
Implementation architecture of the ILBQA on an FPGA platform.

**Figure 8 sensors-22-00175-f008:**
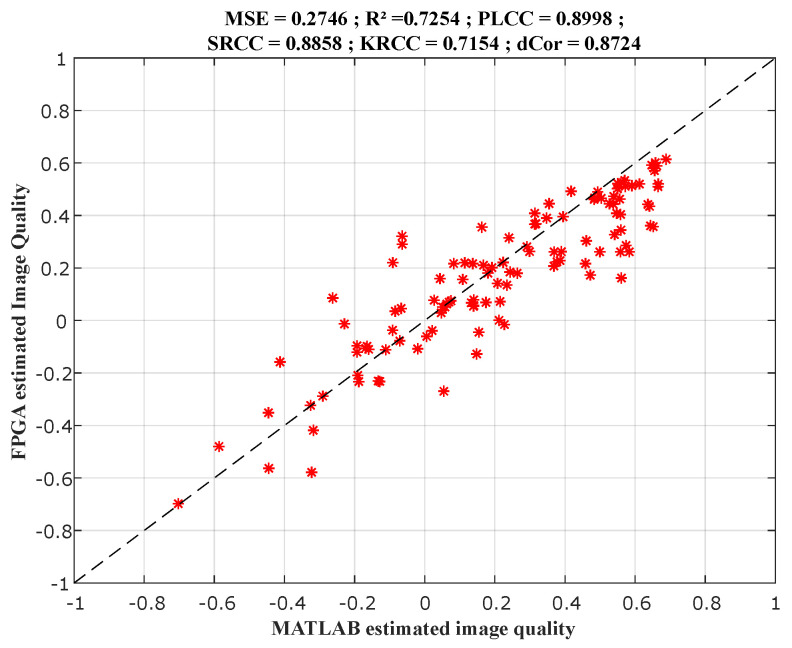
Comparison between MATLAB and FPGA implemented scores.

**Figure 9 sensors-22-00175-f009:**
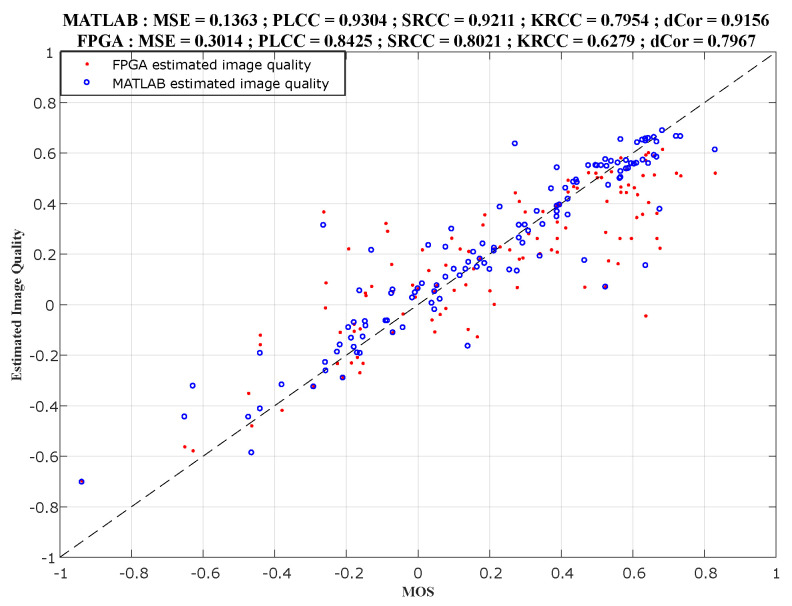
Comparison between MOS, MATLAB, and FPGA implemented scores.

**Figure 10 sensors-22-00175-f010:**
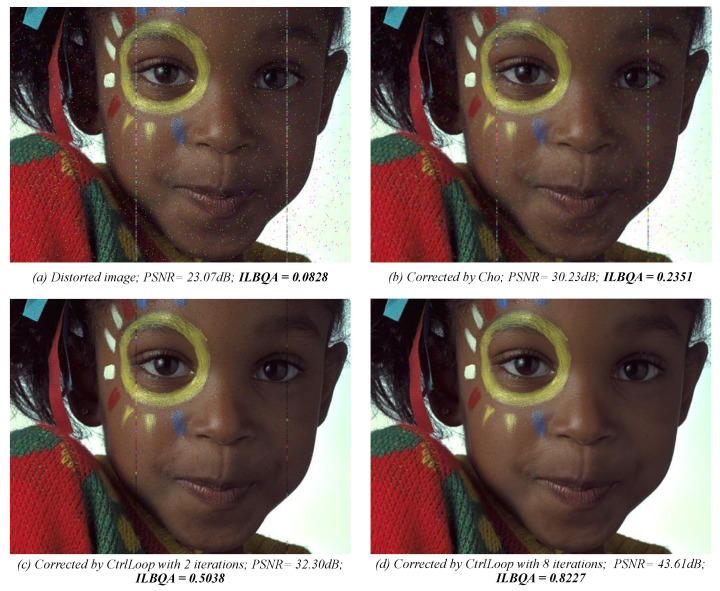
Example of using ILBQA index in a control loop process.

**Table 1 sensors-22-00175-t001:** Extracted features based on NSS in spatial domain.

Feats	Feature Description	Computation Procedure
**f1–f2**	Shape and variance	Fit GGD to MSCN coefficients
**f3–f6**	Shape, mean, left variance, right variance	Fit AGGD to horizontal (H) pairwise
**f7–f10**	Shape, mean, left variance, right variance	Fit AGGD to vertical (V) pairwise
**f11–f14**	Shape, mean, left variance, right variance	Fit AGGD to first diagonal (D1) pairwise
**f15–f18**	Shape, mean, left variance, right variance	Fit AGGD to second diagonal (D2) pairwise

**Table 2 sensors-22-00175-t002:** Extracted features based on GM and LoG.

Feats	Feature Description
**f1–f10**	marginal probability functions of GM
**f11–f20**	marginal probability functions of LoG
**f21–f30**	measure of the overall dependency of GM
**f31–f40**	measure of the overall dependency of LoG

**Table 3 sensors-22-00175-t003:** Extracted features based on SSE.

Feats	Feature Description
**f1–f2**	Mean and skew of spatial entropy values for the first scale
**f3–f4**	Mean and skew of spectral entropy values for the first scale
**f5–f6**	Mean and skew of spatial entropy values for the second scale
**f7–f8**	Mean and skew of spectral entropy values for the second scale
**f9–f10**	Mean and skew of spatial entropy values for the third scale
**f11–f12**	Mean and skew of spectral entropy values for the third scale

**Table 4 sensors-22-00175-t004:** Correlation score of MOS vs. estimated image quality of different image databases using different ML methods for DLBQA.

DB		ANN	SVM	DT	FL
	**PLCC**	0.871	**0.899**	0.878	0.786
**TID**	**SRCC**	0.859	**0.883**	0.862	0.751
	**dCor**	0.850	**0.882**	0.854	0.762
	**PLCC**	0.947	0.960	**0.962**	0.927
**LIVE**	**SRCC**	0.946	0.958	**0.960**	0.936
	**dCor**	0.939	0.950	**0.954**	0.927
	**PLCC**	0.627	**0.801**	0.741	0.673
**NLC**	**SRCC**	0.566	**0.774**	0.739	0.613
	**dCor**	0.596	**0.779**	0.732	0.630

**Table 5 sensors-22-00175-t005:** Spearman’s correlation between intermediary metrics and MOS for different image databases.

DB	IM_	ANN	SVM	DT	FL
	**NSS**	0.647	0.703	**0.842**	0.615
	**GM**	0.697	0.597	**0.849**	0.629
**TID**	**LOG**	0.547	0.463	**0.780**	0.498
	**SSE**	0.623	**0.865**	0.856	0.431
	**NSS**	0.931	**0.939**	0.930	0.915
	**GM**	0.946	0.924	**0.954**	0.940
**LIVE**	**LOG**	0.942	0.928	**0.954**	0.937
	**SSE**	0.868	**0.954**	0.937	0.632
	**NSS**	0.603	0.618	**0.717**	0.569
	**GM**	0.441	0.488	**0.685**	0.499
**NLC**	**LOG**	0.527	0.529	**0.741**	0.552
	**SSE**	0.444	**0.763**	0.683	0.440

**Table 6 sensors-22-00175-t006:** Brownian’s distance between the group of intermediary metrics and MOS.

dCor	TID2013	LIVE	Live Challenge
**ANN**	0.745	0.951	0.632
**SVM**	0.835	0.957	0.772
**DT**	**0.907**	**0.969**	**0.823**
**FL**	0.691	0.937	0.634

**Table 7 sensors-22-00175-t007:** Correlation score of MOS vs. estimated image quality of different image databases using different ML methods for ILBQA.

DB		ANN	SVM	DT	FL
	**PLCC**	0.925	0.936	**0.954**	0.924
**TID**	**SRCC**	0.912	0.924	**0.946**	0.913
	**dCor**	0.908	0.919	**0.942**	0.908
	**PLCC**	0.971	0.978	**0.982**	0.970
**LIVE**	**SRCC**	0.970	0.977	**0.979**	0.968
	**dCor**	0.961	0.971	**0.976**	0.961
	**PLCC**	0.818	0.815	**0.894**	0.826
**NLC**	**SRCC**	0.777	0.848	**0.877**	0.793
	**dCor**	0.792	0.851	**0.879**	0.802

**Table 8 sensors-22-00175-t008:** Performance comparison on three benchmark IQA databases.

Methods	LIVE		TID2013		LIVE-CH	
	SRCC	PLCC	SRCC	PLCC	SRCC	PLCC
DIQaM [33]	0.960	0.972	0.835	0.855	0.606	0.601
BIECON [48]	0.958	0.960	0.717	0.762	0.595	0.613
DIQA [32]	**0.975**	**0.977**	0.825	0.850	0.703	0.704
DB-CNN [49]	0.968	0.971	0.816	0.865	**0.851**	**0.869**
AIGQA [16]	0.960	0.957	**0.871**	**0.893**	0.751	0.761
**CNN-BQA (proposed)**	0.961	0.966	0.881	0.876	0.772	0.786
DIIVINE [26]	0.925	0.923	0.654	0.549	0.546	0.568
BRISQUE [28]	0.939	0.942	0.573	0.651	0.607	0.585
ILNIQE [27]	0.902	0.865	0.519	0.640	0.430	0.510
FRIQUE [31]	0.948	0.962	0.669	0.704	0.720	0.720
SSEQ [30]	0.941	0.943	0.883	0.886	0.772	0.807
GMLOGQA [29]	0.956	0.970	0.883	0.877	0.782	0.805
**DLBQA (proposed)**	0.951	0.957	0.892	0.893	0.779	0.862
**ILBQA (proposed)**	**0.977**	**0.983**	**0.934**	**0.942**	**0.876**	**0.883**

**Table 9 sensors-22-00175-t009:** Percentage of time consumed by each feature extraction axes.

Axes	Percentage of Time (%)
**NSS**	7.98
**GM**	3.10
**LOG**	3.10
**SSE**	85.77

**Table 10 sensors-22-00175-t010:** Comparison of time complexity for different BIQA approaches.

NR-IQA	Time (second)
BLIIND-II	76.12
DIIVINE	25.40
BRISQUE	0.168
GMLOGQA	0.131
SSEQ	1.807
**ILBQA**	2.126

**Table 11 sensors-22-00175-t011:** Device performance utilization estimates in HLS synthesis phase.

	NSS	GM-LoG	SSE	DT	FL	ILBQA
**clk est. (ns)**	21.79	21.84	21.84	11.31	16.82	21.85
**Min clk.**	79,302	1648	570	2	42	80,370
**Max clk.**	9,400,263	2,909,751	25,231,749	17	43	84,720,454
**BRAM**	1235	1 052	554	0	0	2328
**DSP**	394	436	141	0	96	1136
**FF**	59,097	69,705	20,872	10,936	5245	167,801
**LUT**	181,656	247,775	50,903	128,122	11,779	497,622

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
