# Peer review of "Visual Perceptual Quality Assessment Based on Blind Machine Learning Techniques"

_sensors, 2021, doi:10.3390/s22010175_

Round 1

Reviewer 1 Report

Pages 5, 6; Equations:

You use »where« and »Where«. Probably, it is better to end the equations with the comma, and continue with »where«.

Pages 5, 6; Tables 2, 3 and 4; Page 7, Line 177:

You described 18 features based on NSS (Table 1), 40 features based on GM and LoG (Table 2), 12 features based on SSE (Table 3). Overall number of features is (18 + 40 + 12 = 70). Later, on line 177, you mention:

“In the feature extraction phase, 88 features were extracted in the images.”

You did not explain clearly how other 18 missing features are determined? You should also describe these missing features in similar way than other 70 features.

Page 8; Line 190:

You remove strongly correlated features, superfluous features. Are more like to be two features correlated within the group of features (for example, within NSS based) or can be features correlated between different groups (for example, features between NSS and SSE group)?

Author Response

We would like to thank you for the constructive comments and suggestions. We have responded to all points raised in the reviews in the rebuttal and we have modified and extended our manuscript accordingly. For convenience, the changes in the revised manuscript with the respect to the original manuscript are highlighted. The changes are also clearly indicated in the rebuttal.

Reviewer 2 Report

In this paper, a very interesting method for quality assessment in images is proposed. The explanation of the method is really clear, and the experiments carried out are extensive and well designed. A lot of comparisons with old methods have been performed including execution times.

Some minor comments:

Just a few references are from the last year so it would be interesting to add more recent work. Machine learning is a very active field, so it is important to have an updated state of the art.

Figure 2 should be bigger or with a bigger font.

There are no future works in conclusion. You should add them

Author Response

(The authors gave the same response as above.)

Reviewer 3 Report

The research developed is interesting, although in my view the article should be organised in an alternative way to improve its explanatory power. Some suggestions for authors would be as follows:

  1. In general, figures and tables should be relocated to the corresponding parts where they are referenced whenever possible.
  2. The resolution of Figures 2 and 3 should be improved.
  3. In line 90, when talking about the state of the art, it should be specified which algorithms the authors are referring to or indicate the section where this is done. (Section 4.1?)
  4. The inclusion of information that is easy to deduce or understand from the context or subject matter would improve the readability and explainability of the text. For example, in line 103 when entering I(i,j) it could be specified which information is stored by each of the picture elements under consideration.
  5. In general, and this is in my opinion the major weakness of the present work, the introduction of the methods should be done in a more complete and explanatory way in cases where results related to the proposed methodology are involved. As an example (but not the only one), the development of the equations (7), (8), etc..
  6. Table 1 shows 'special' instead of 'spatial'.
  7. In table 4 the acronyms PLCC, SRCC, dCor are used without having been previously introduced.
  8. The measures 'Spearman correlation' and 'brownian distance' have not been introduced before and their use has not been justified.
  9. The measures 'Spearman correlation' and 'brownian distance' have not been introduced before and their use has not been justified.
  10. Line 330: It is understood that parallel computing is not being used, but there may be room for doubt in the current wording.
  11.  

Author Response

(The authors gave the same response as above.)

Round 2

Reviewer 3 Report

Thanks to the authors for addressing the suggestions made in the previous round of review. In my opinion, the article has gained in clarity. A minor consideration that could still be considered would be:

  1. The placement of figures 7, 8 and table 11 may lead to confusion.
  2. More detail on the development and derivation of the equations in section 2 may be of interest.

However, the comments indicated are minor and it could therefore be published in its present form. Therefore, although I recommend a minor revision, I leave it up to the authors to carry it out. 

Author Response

Thanks to the reviewer for these constructive comments and suggestions. For convenience, the changes in the revised manuscript with the respect to the original manuscript are highlighted.

  1. We have moved Figures 7,8 and Table 11 where possible in the Session where they are cited..
  2. more details on the equations of Session 2 are given in the referenced documents in this Session.